# A Study on the Summer Microclimate Environment of Public Space and Pedestrian Commercial Streets in Regions with Hot Summers and Cold Winters

**Junyou Liu** [1] , **Haifang Tang** [2], **Bohong Zheng** [1,*] **and Zhaoqian Sun** [1]

[1] School of Architecture and Art, Central South University, Changsha 410083, China
[2] Institute of Territory Spatial Planning, China Machinery International Engineering Design & Research Institute Co., Ltd., Changsha 410007, China
* Correspondence: zhengbohong@csu.edu.cn

**Featured Application: This study can guide urban planners and urban designers to think about how to improve thermal comfort by optimizing street flow lines, the locations of open space, and greening arrangements in pedestrian commercial streets, especially those in the hot summer and cold winter regions of China.**

**Abstract:** Pedestrian commercial streets are an important part of a city. However, the open outdoor street is easily affected by the external climate, and a poor microclimate environment can indirectly affect the volume of visitors to the commercial street. This paper takes pedestrian commercial streets in regions with hot summers and cold winters as the research object in order to obtain reasonable prototypes of street space. Adopting the experimental method of controlling variables, microclimate simulation analysis is conducted on different street flow lines, various locations of open space, and the different greening arrangements of typical street spaces. This paper also proposes design strategies for improving the microclimate environment, such as reserving ventilation passages in the dominant wind direction, setting up air buffer areas to increase the "wind storage" effect, building an open space in the upwind direction to increase the "wind absorption" effect, preventing planar greening space from hindering airflow in streets with poor ventilation, and establishing planar green space in the upwind direction to increase the coverage of the cooling effect of plants. In this paper, comfort in the outdoor microclimate comfort is taken into consideration in commercial street design, aiming to achieve the revitalization of commercial streets through "micro renovation" and provide some reference for the future design of commercial streets.

**Keywords:** public space; commercial street; outdoor microclimate; PET

## 1. Introduction

In the past, traditional pedestrian commercial streets were the main place for citizens to shop. However, in recent years, online shopping at home has become increasingly popular, and the traditional activities that took place on pedestrian commercial streets have been greatly impacted [1]. The reason behind this is not only the many advantages of online shopping, such as its time-saving potential, convenience and variety of choice, but also some disadvantages that are associated with traditional pedestrian commercial streets. One of the main disadvantages of traditional pedestrian commercial streets is their poor level of outdoor comfort, as they are mainly hot in the summer and experience cold wind in the winter [2–4]. In contrast, online shopping is much more comfortable.

As an important window into the image of a city, pedestrian commercial streets reflect the vitality of the entire city [5]. Sun et al. (2020) explored the interface pattern of 14 pedestrian commercial streets in Beijing, Shanghai, and Xuzhou, China, using related theories of typology; they summarized that the plane form of related pedestrian streets

could be divided into straight, curved, concave, and broken-line. The facade form of pedestrian streets can be divided into uniform height type, diversified height type, and tower type. The profile morphology can be divided into straight and indented [6]. Analysis of the similarities among various commercial pedestrian streets can help to give us a better understanding of their related features. On this basis, they were transformed in order to optimize the spatial structure of pedestrian commercial streets. In the process of renewing pedestrian commercial streets in China, designers paid attention to the protection of the original texture and structure of the block, and to increasing the vitality of pedestrian commercial streets [6–8]. Due to the protection of the texture of pedestrian commercial streets, the historical style of many commercial streets was preserved. The traditional design and renovation of pedestrian commercial streets usually focus on the traffic flow lines and image of the city, but pay little attention to the thermal comfort of outdoor public spaces, which can affect the volume of visitors to commercial streets. Therefore, it is necessary to improve the outdoor comfort of pedestrian commercial streets so as to attract more visitors and avoid the loss of vitality on urban pedestrian commercial streets.

In terms of research on the outdoor microclimate, scholars from various countries have conducted detailed analysis on the impact factors of the outdoor microclimate from different perspectives and for different climatic regions. For example, Ali-Toudert and Mayer (2005) conducted a simulation study on the impact of the aspect ratio and orientation of an urban street canyon on outdoor thermal comfort; this study was based on the building and street characteristics and microclimate conditions of Ghardaia, Algeria, a city in a dry and hot region, and was performed using ENVI-met microscale meteorological simulation software. The research results showed that the depth of the buildings, the width of the buildings along the street, and the depth/width ratio of the street are the main factors that affect street temperature [9,10]. Johansson (2004) studied the urban geometry and microclimate at street level in the city of Fez, which is located in the interior of Morocco and has a typical dry and hot climate; they found that the deep canyon was considerably cooler than the shallow canyon [11,12]. Bourbia and Boucheriba (2010) conducted research on the impact of road design in the downtown area of Constantine City, located in a semi-arid region in Algeria, on the urban microclimate. They found that the depth/width ratio of the street and the opening factor of the sky are the two main factors that affect the road surface temperature [13]. Guo (2020) categorized block morphological types in three cities located in the cold regions of China (Shijiazhuang, Zhengzhou, and Xi'an), and analyzed the differences in the microclimate among different block spatial morphologies and the reasons for these differences. Eight quantifiable indicators related to block morphology were considered, including the average building height, building density, building functional mix, plot ratio, average street height/width ratio, average sky openness, average windward area ratio, and greenery ratio. He found, in cold regions, that the depth/width ratio of the street, the materials comprising the underlying surface, the relationship between the street and the dominant wind direction, and the building facades on both sides of the street are the main factors that affect the thermal environment of the street space [14,15]. Cui (2020) explored the association between commercial block morphological elements and human thermal comfort in Harbin, China, which is located in a severely cold region. He found that in severely cold regions, the directional depth-width ratio of the commercial street, building density, plot ratio, greening rate and building layout are the main factors that affect outdoor thermal comfort. By analyzing several representative studies, it is evident that they mainly focus on the influence that different distributions of morphological street elements have on the microclimate, with a focus on the correlation between buildings, the relationship between buildings and streets, and the relationship between greening and the microclimate [16].

According to the Chinese national Uniform Standard for Design of Civil Buildings (GB50352-2019) [17], China is divided into seven different climate zones. The division of climate zones is conducive to the better design of buildings for energy efficiency in different climate regions. The hot summer and cold winter region is one of the seven

main building climate zones in China [17,18]. Some researchers have conducted studies on how to optimize urban spatial patterns in hot summer and cold winter regions in order to improve thermal comfort. For example, Cao et al. (2022) explored the impact of trees on the thermal comfort of sidewalks in Shanghai, China, and found that green coverage has a positive effect on improving thermal comfort in summer, with varying effects under different road orientations and aspect ratios [19]. Li et al. (2022) studied the relationship between the outdoor environment and thermal comfort in old residential neighborhoods in Hangzhou, China, a city located in a hot summer and cold winter region; they found that factors such as size, greening, layout, water systems, and location have a significant impact on thermal comfort [20]. Huang et al. (2020) conducted measurement studies on a site within the campus of the Southwest University of Science and Technology in Mianyang, China, a city located in a hot summer and cold winter region, and explored the thermal comfort of different ground materials. The study found that ground materials have a significant impact on the surface temperature in winter, and on the relative humidity and PET (predicted mean vote) in summer [21]. Ma et al. (2019) took the Dao He Old Block in Taizhou, China, where it is hot in summer and cold in winter, as an example, explored the thermal comfort of pedestrian areas, and found that architectural shadow became an important factor in improving the thermal comfort of street canyons during the day as the height of buildings on both sides of the road gradually increased [22]. Cao et al. (2022) also pointed out that the bigger the aspect ratio (the ratio of the height of buildings on both sides of the street to the width of the street) was, the taller the buildings were. The architectural shadow can be extended and the coverage can be made wider. This also indicates that the thermal comfort of the street can be improved by blocking sunlight with architectural shadow during the day in summer for buildings on both sides of the road. The wider the coverage of the architectural shadow is, the wider the range of the thermal comfort offered by it is [19]. By reviewing the relevant literature, it is clear that improving the spatial form of urban blocks with the aim of enhancing their thermal comfort has drawn the attention of some researchers. The region in China with hot summers and cold winters has high temperatures in summer and low temperatures in winter, so it is very important to improve the thermal comfort of the living environment in this region. Therefore, the study on the spatial form of pedestrian commercial streets in China's hot summer and cold winter regions and the exploration of how to optimize commercial blocks from the perspective of spatial form will undoubtedly help to improve the thermal comfort of pedestrian commercial streets and increase their attractiveness. However, there is limited research on the thermal comfort of pedestrian commercial streets in hot summer and cold winter regions. At present, urban construction in China is relatively complete. In the next stage, the focus of construction will be on the "micro renovation" of blocks, and pedestrian commercial streets with a high density, high flow, and high requirements for outdoor comfort should be given the top priority. Researchers believe that during the updating process, the focus should be on public spaces and streets because people spend much time in outdoor public spaces. Therefore, this paper analyzes the impact of the morphology of public spaces on the microclimate of pedestrian commercial streets in regions with hot summers and cold winters, with the aim of providing strategies for improving the microclimate of pedestrian commercial streets and achieving the revitalization of traditional commercial streets. Considering the reality of pedestrian commercial streets in hot summer and cold winter zones in China, this paper explores improvements in the spatial form of pedestrian commercial streets in order to achieve thermal comfort by optimizing street flow lines, the locations of open space, and the greening arrangements of pedestrian commercial streets. It points out the importance of our research in the light of this situation.

## 2. Materials and Methods

### 2.1. Research Object

Regions with hot summers and cold winters in China are very hot in summer and very cold in winter, with high humidity all year round and a poor comfort level compared

with other regions at the same latitude. Compared to the cold and humid winter, the hot and sultry summer makes people feel more uncomfortable, and fewer people engage in outdoor activities in summer than in the other three seasons. The summer climate has a greater impact on the volume of visitors to commercial blocks, so summer is selected as the research season in this paper.

According to the division of climate zones in the Chinese Uniform Standard for Design of Civil Buildings (GB50352-2019) [17], the regions with hot summers and cold winters include the whole area of Shanghai, Chongqing, Hubei, Hunan, Jiangxi, Anhui and Zhejiang, the eastern half of Sichuan and Guizhou, the southern half of Jiangsu and Henan, the northern half of Fujian, the southern part of Shaanxi and Gansu, and the northern part of Guangdong and Guangxi (Figure 1) [18]. In this paper, seven typical commercial pedestrian streets, namely Jiefangbei Pedestrian Street in Chongqing, Nanjing Road Pedestrian Street in Shanghai, Huangxing South Road Pedestrian Street in Changsha, Hunan Province, Jianghan Road Pedestrian Street in Wuhan, Hubei Province, Huaihe Road Pedestrian Street in Hefei, Anhui Province, Xingguang Avenue Pedestrian Street in Hangzhou, Zhejiang Province, and Shengli Road Pedestrian Street in Nanchang, Jiangxi Province, are selected as research objects. The geographical locations of the seven cities mentioned above are marked by red stars in Figure 1 below.

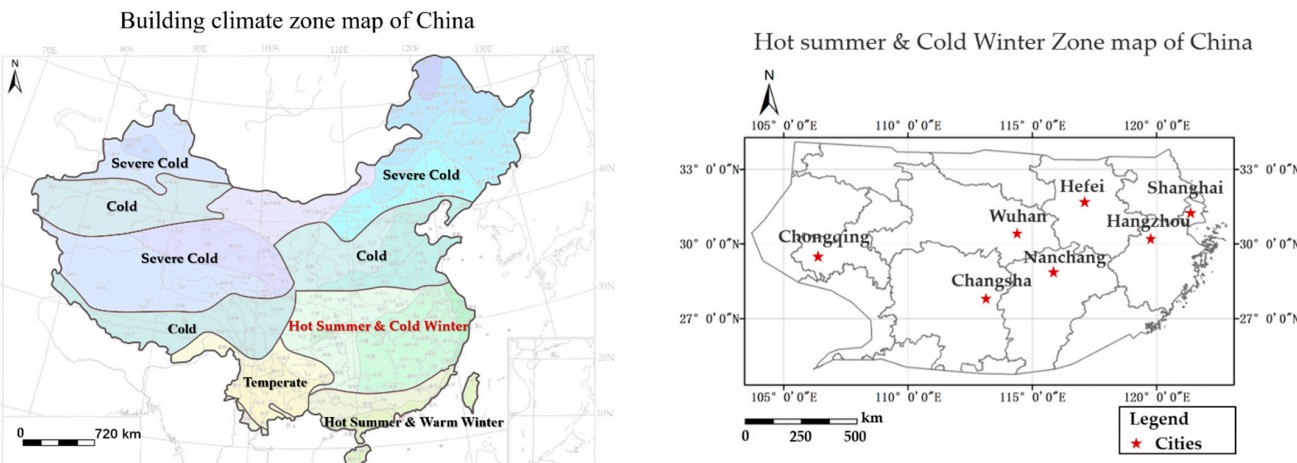

**Figure 1.** Distribution of the studied pedestrian commercial streets.

## 2.2. Research Methods and Experimental Parameter Design

### 2.2.1. Research Methods and Indicators

Various research methods have been used to study the outdoor microclimate environment in the existing research, such as directly obtaining meteorological data through actual measurements in order to compare the advantages and disadvantages of the microclimate in different situations, establishing mathematical models to study the correlation between various research factors and the microclimate, and determining the improvement effects of different research factors on the microclimate through the numerical simulation of ideal models. By comprehensively comparing the advantages and disadvantages of these methods, it is clear that the numerical simulation of ideal models is more suitable and is thus now a common method in the study of microclimates, since it not only eliminates the impact of other factors on the experiment, but also conducts multi-factor analysis.

Since the 20th century, there have been more than 160 evaluation indicators used to describe the sensory comfort of the human body [23]. Through the summary and comparison of multiple indicators by many scholars around the world, it has been found that the physiological equivalent temperature (PET) [24,25], predicted mean vote (PMV) [26], and standard effective temperature (SET*) [27] are the indexes of microclimate comfort that can better describe the outdoor microclimate environment and that are more convenient to study. Among them, the PET is the comfort index with the highest frequency of use [28,29].

Chinese researchers have obtained the PET for outdoor microclimate analysis through questionnaires [30,31], Rayman calculations [32], and ENVI-met simulations [33,34]. The final research results show that the results of the outdoor microclimate analysis using the PET obtained via these methods are relatively accurate, and the most commonly used method in recent years is ENVI-met simulation.

The PET is a comfort evaluation index derived on the basis of the Munich Energy Balance Model for Individuals (MEMI), with a comprehensive consideration of the impact of various major meteorological indexes (such as temperature, humidity and wind speed), human activities, human clothing, and personal body parameters on environmental comfort [35]. It can also be considered as different combinations of various microclimate factors (mainly temperature, humidity, and wind speed) that the human body can tolerate at the same time [36]. Figure 2 below shows the relationship between the PET and human senses.

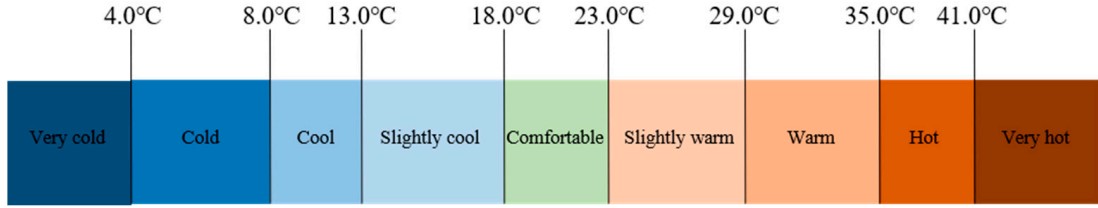

**Figure 2.** Relationship between PET and human senses.

### 2.2.2. Experimental Parameter Design

The simulation date was chosen as 12 July 2022. Considering that the temperature is generally high around 2:00 p.m. in hot summer and cold winter regions, this research attaches great importance to the comparison of the simulation results of thermal comfort of different models at 2:00 p.m. Based on the data downloaded from the China Meteorological Data Service Center, the initial simulation conditions were set as shown in Table 1.

**Table 1.** Initial simulation conditions.

| Initial Simulation Time | Simulation Duration | Temperature (°C) | Relative Humidity (%) | Average Wind Speed (m/s) | Average Wind Direction (°) |
|---|---|---|---|---|---|
| 11:00 | 10 h (From 11:00 to 20:00) | 33.3 (11:00)<br>33.8 (12:00)<br>34.9 (13:00)<br>35.4 (14:00)<br>35.3 (15:00)<br>35.0 (16:00)<br>34.6 (17:00)<br>34.0 (18:00)<br>33.1 (19:00)<br>32.2 (20:00) | 59 (11:00)<br>58 (12:00)<br>56 (13:00)<br>58 (14:00)<br>57 (15:00)<br>56 (16:00)<br>59 (17:00)<br>64 (18:00)<br>59 (19:00)<br>66 (20:00) | 3.45 | 175.62 |

### 2.2.3. Verification of ENVI-Met Feasibility

The Version 5.1.1 of the ENVI-met was used to carry out the simulations. Many scholars have proven that the research results regarding the microclimate at the block level, which use the PET simulated by ENVI-met software, are reliable. However, in order to better find the errors in the simulation experiment and prove that using ENVI-met is feasible as the simulation software for this research, the feasibility of ENVI-met software is to be verified in this paper. The numerical significance of the PET is its ability to reach the temperature corresponding to the same thermal state as the outdoor environment in a typical indoor environment with a temperature of 20 °C and a relative humidity of 50% [37,38]. Therefore, determining whether the PET value simulated by ENVI-met is reasonable is approximately equivalent to determining whether the temperature and humidity calculated by ENVI-met are within the error range.

The researchers conducted field data collection at a fixed location on Pozi Street in Changsha City from 11:00 a.m. to 8:00 p.m. on 12 July 2022. The selected instrument was a JT2020 multi-function tester (by China Shiji Jiantong Technology Co., Ltd., Beijing, China). The range and accuracy of the instrument are shown in Table 2 below. The error in the simulation was obtained by comparing the measured and simulated data for temperature and humidity at the observation points between 11:00 a.m. and 8:00 p.m.

**Table 2.** Range and accuracy of the measuring instrument.

| Meteorological Parameters | Range | Accuracy |
|---|---|---|
| Temperature | From −20 °C to 125 °C | ±0.5 °C |
| Relative humidity | Between 0 and 100% RH | ±3% RH |
| Wind speed | From 0.05 to 5 m/s | 0.05~2.0 m/s ± (0.05 + 2% reading) 2.0~5.0 m/s ± (0.1 + 2% reading) |

As shown in the table below, the temperatures of a fixed point within the study area were measured and simulated every hour, on the hour, from 11 o'clock to 20 o'clock on 12 July 2022. After analyzing the temperatures, it is clear that the coefficient of determination ($R^2$) between the measured temperature and simulated temperature was 0.89, that the root mean square error (RMSE) was 1.41, and that the index of agreement (d) was 0.71. For the measured humidity and simulated humidity, the R2 was 0.81, the RMSE was 2.7%, and d was 0.76. Compared with other similar studies, this simulation study is in a reasonable error range [39]. The measured temperature and humidity and their corresponding simulated temperature and humidity are shown in Table 3 below.

**Table 3.** Comparison between measured data and simulated data.

| Time | Measured Temperature (°C) | Simulated Temperature (°C) | Measured Relative Humidity (%) | Simulated Relative Humidity (%) |
|---|---|---|---|---|
| 11:00 | 33.7 | 32.6 | 57.4 | 54.3 |
| 12:00 | 34.1 | 33.3 | 54.6 | 52.6 |
| 13:00 | 35.9 | 34.0 | 50.4 | 51.0 |
| 14:00 | 37.8 | 34.6 | 50.0 | 51.2 |
| 15:00 | 37.3 | 34.8 | 53.1 | 50.8 |
| 16:00 | 37.2 | 34.7 | 50.2 | 50.3 |
| 17:00 | 36.0 | 34.4 | 53.1 | 50.3 |
| 18:00 | 35.4 | 33.8 | 55.8 | 51.9 |
| 19:00 | 34.9 | 33.0 | 58.8 | 54.7 |
| 20:00 | 33.5 | 32.4 | 59.5 | 55.9 |

### 2.3. Prototype Extraction of Street Space

The main research object of this paper was the morphology of the public space of streets in commercial blocks. When extracting the texture prototype of the block space, the morphologies of the buildings were not analyzed. By analyzing the relationships among the textures of seven typical pedestrian streets, it was found that the morphological elements of the public space of pedestrian commercial streets mainly include the moving lines of streets, the locations of open space, and the greening layout. There are two main types of moving lines of streets, namely the single-line type and multi-line type. According to their relationship with the dominant wind direction, single-line-type streets are roughly divided into streets that are perpendicular to the dominant wind direction and streets

that are parallel to the dominant wind direction, while multi-line-type streets include both streets that are perpendicular to the dominant wind direction and streets that are parallel to the dominant wind direction. The location of the open space is usually at the entrance or the middle of the commercial street, or at a random mixed position. There are four types of greening layout: point, line, surface, and mixture. A delicate and beautiful pedestrian street landscape is formed through different greening arrangements [40–42]. Figure 3 shows the prototype extraction of the morphological elements of a typical commercial street. In Figure 3, the red circle represents the open space located at both ends of the street, and the blue circle represents the open space not located at both ends of the street.

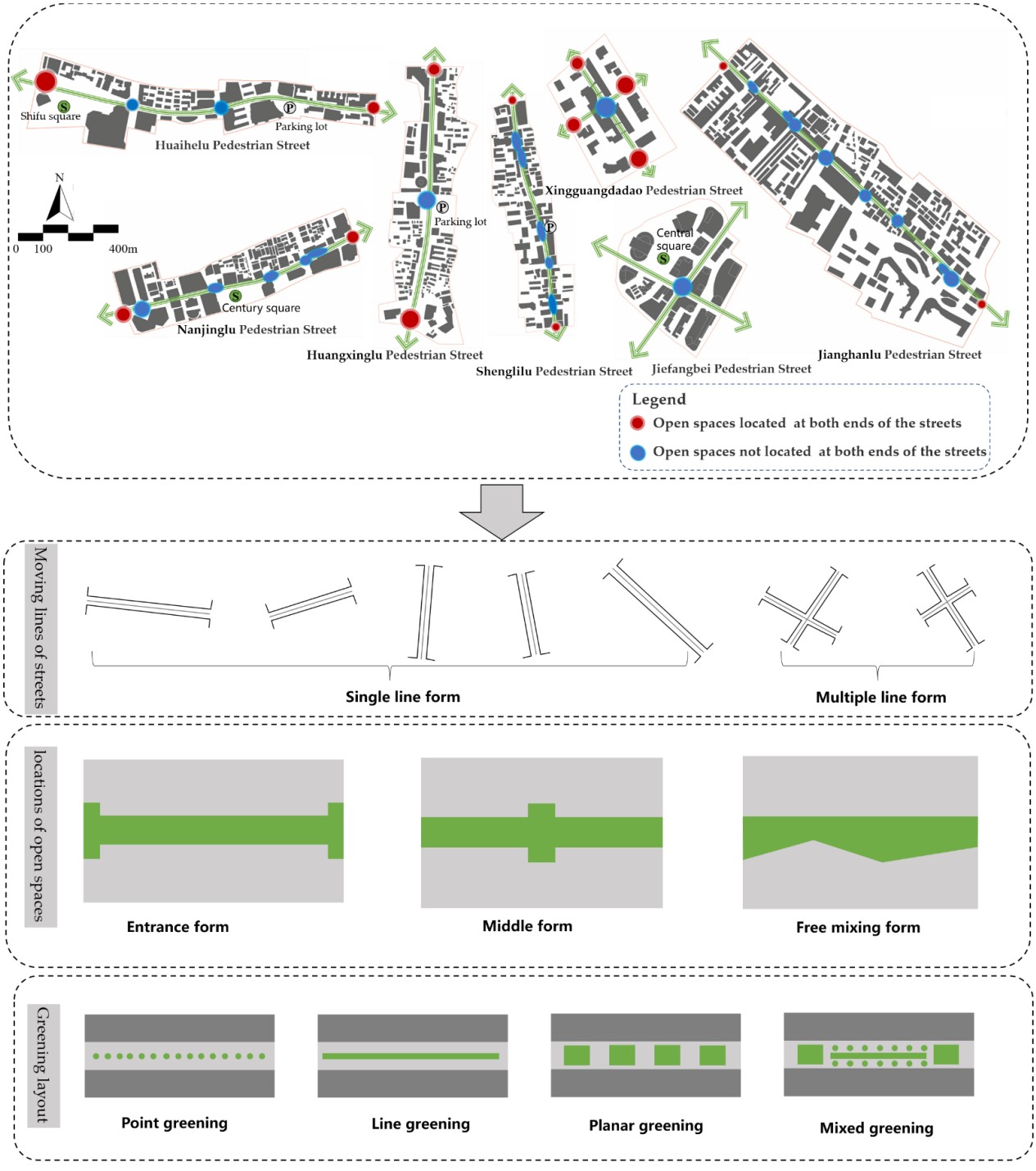

**Figure 3.** Prototype extraction of morphological elements of typical commercial street.

After the prototype of the street space was extracted, the microclimate in the summer environment of the public space of pedestrian commercial streets in regions with hot summers and cold winters was studied in this paper from the perspective of the moving lines of streets, locations of open space, and greening layout.

### 2.4. Experiment Design

2.4.1. Moving Lines of Streets

In order to study the microclimate environment characteristics of the different moving lines of streets, three models were designed in Study 1 in order to reduce the impact of other factors on the experiment, which can cause experimental errors. The three models have similar micro-environment conditions. The specific micro-environment conditions are shown in Table 4.

**Table 4.** Micro-environment conditions of the models in Study 1.

| Land Area | Plot Ratio | Building Density | Greening Rate | Greening Structure | Number of Macrophanerophytes |
|---|---|---|---|---|---|
| 1 ha | 1.2 | 36% | 30% | Macrophanerophytes—herbs | 50 |

The land used in Model 1 was 160 M × 62.5 M in size, and the street direction was east–west, perpendicular to the average wind direction during the study period. The moving line of the street was a single-line type, and the greening was parallel to the building layout and was a combination of linear green space and planar green space. The land used in Model 2 was 62.5 M × 160 M in size. The design of Model 2 was based on Model 1, with a clockwise rotation of 90°, so that the street direction was parallel to the average wind direction on the study day. In Model 3, the land used was 100 M × 100 M in size, with both an east–west street and north–south street. The moving lines of the streets were a multi-line type in a "cross" shape. The greening pattern included green space surrounding buildings and buildings surrounding green space. The greening layout was a combination of linear green space and planar green space. The specific design of the three models is shown in Figures 4 and 5.

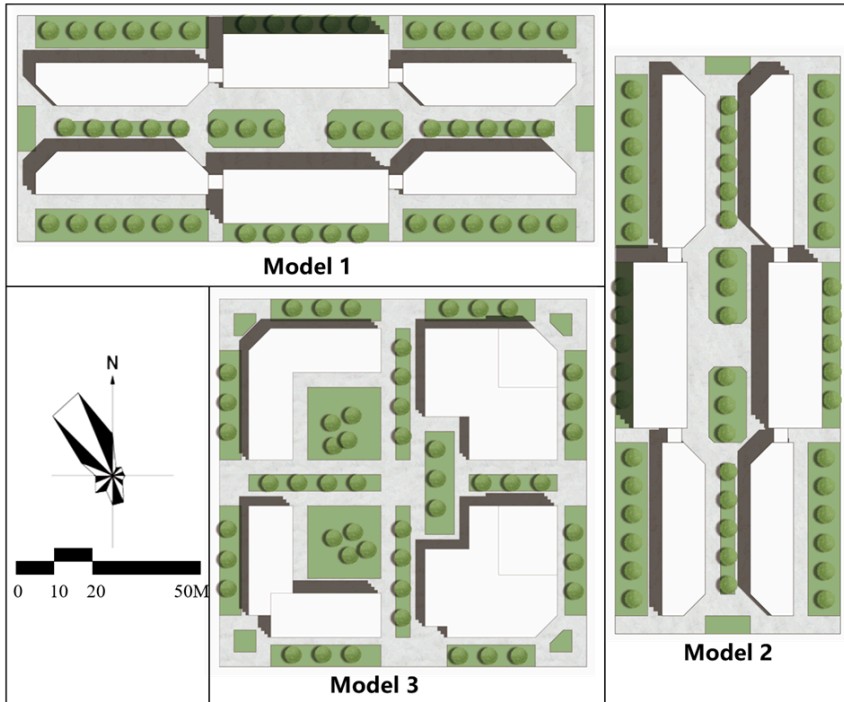

**Figure 4.** Design of experiment for moving lines of streets.

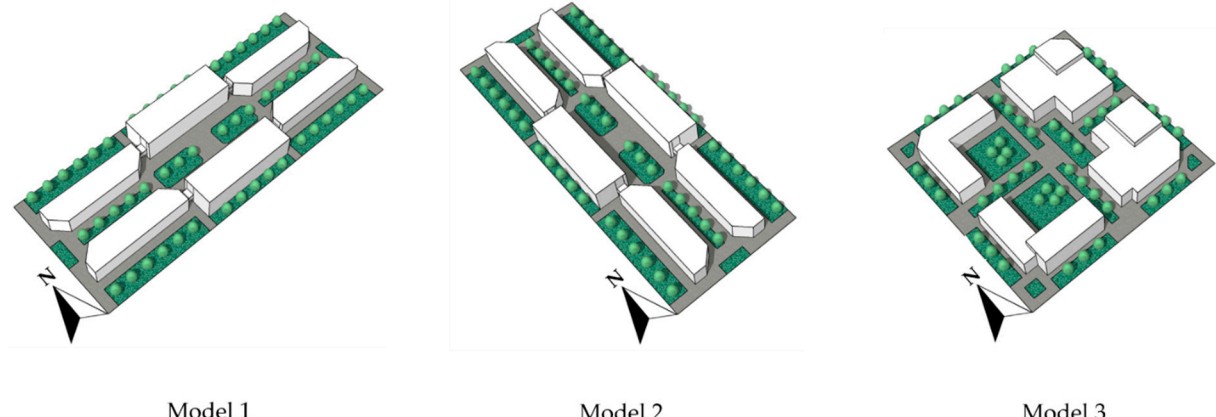

Model 1 Model 2 Model 3

**Figure 5.** Three-dimensional diagram of each model in study 1.

2.4.2. Locations of Open Space

Due to limits on land use under realistic circumstances, there are both commercial streets parallel to the dominant wind direction and commercial streets vertical to the dominant wind direction. In order to reduce the one-sidedness of the study 2 and comprehensively study the microclimate environment characteristics at the different locations of open space, the experimental settings of the open space were divided into two groups in study 2. Group A was based on the east–west street model, with a model size of 160 M × 62.5 M. The open space was designed to be set at the entrance of the commercial street in Model 1, in the middle of the commercial street in Model 2, and at a random position in Model 3, and the characteristics of the microclimate environment were studied under these three conditions. Group B was based on the north–south street model, with a model size of 62.5 M × 160 M. The open space was designed to be set at the entrance of the commercial street in Model 1, in the middle of the commercial street in Model 2, and at a random position in Model 3, and the characteristics of the microclimate environment were studied under these three conditions. The micro-environment conditions of the above six models were similar (Table 5), and the specific designs are shown in Figures 6 and 7.

**Table 5.** Micro-environment conditions of models in Study 2.

| Land Area | Plot Ratio | Building Density | Greening Rate | Greening Structure | Number of Macrophanerophytes |
|---|---|---|---|---|---|
| 1 ha | 1.17 | 39% | 22.8% | Macrophanerophytes—grasses | 48 |

2.4.3. Greening Layout

In order to study the microclimate environment characteristics of different greening layouts, the experimental settings of the greening layout were divided into two groups in study 3. The first group of experiments was based on the east–west street model, with a model size of 160 M × 62.5 M. The greening layout was designed to be a point pattern in Model 1, a linear pattern in Model 2, a planar pattern in Model 3, and a mixed pattern of point, line, and plane in Model 4, and the microclimate environment characteristics under these four conditions were studied. The second group of experiments was based on the north–south street model, with a model size of 62.5 M × 160 M. The greening layout was designed to be a point pattern in Model 1, a linear pattern in Model 2, a planar pattern in Model 3, and a mixed pattern of point, line, and plane in Model 4, and the microclimate environment characteristics under these four conditions were studied. The microclimate conditions of the eight models were similar (Table 6), and the specific designs are shown in Figures 8 and 9.

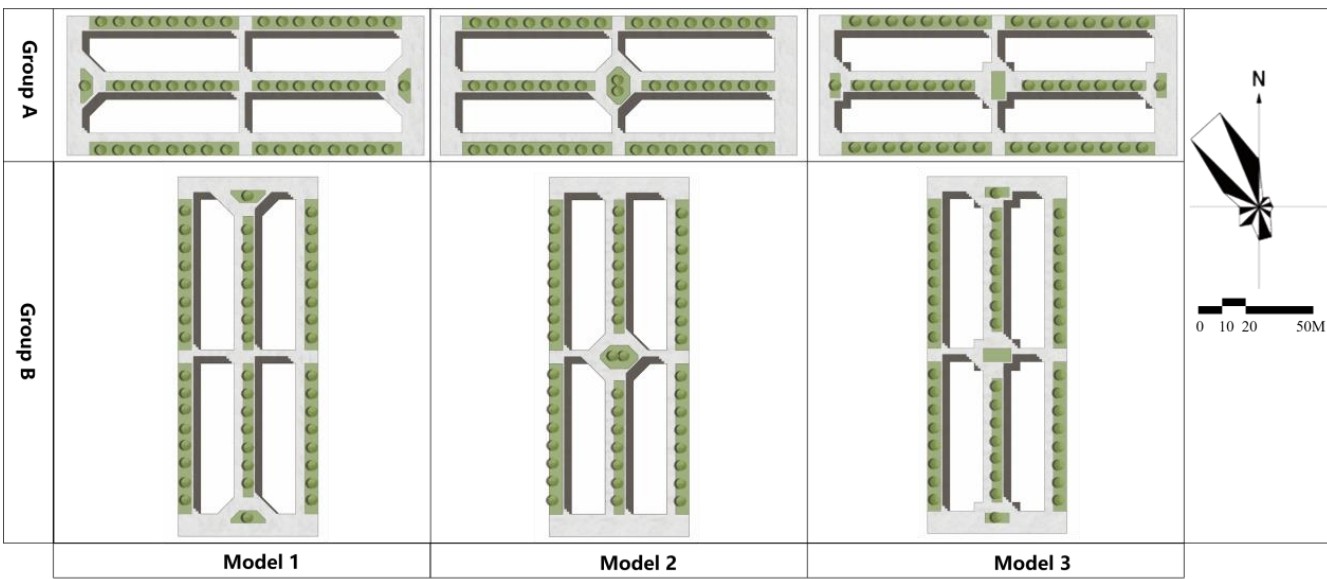

**Figure 6.** Design of experiment for open space.

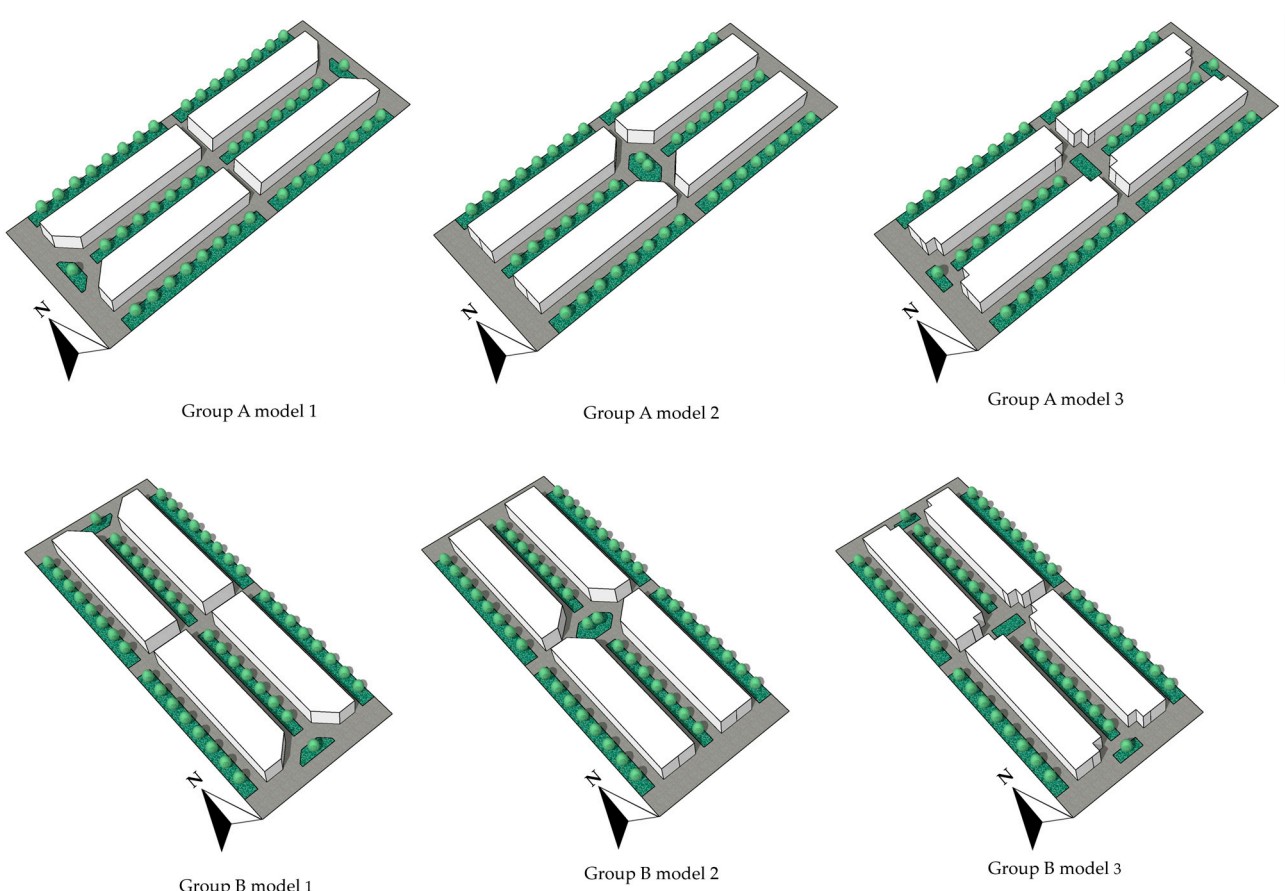

**Figure 7.** Three-dimensional diagram of each model in study 2.

**Table 6.** Micro-environment conditions of models in Study 3.

| Land Area | Plot Ratio | Building Density | Greening Rate | Greening Structure | Number of Macrophanerophytes |
|---|---|---|---|---|---|
| 1 ha | 1.17 | 39% | 19.69% | Macrophanerophytes—grasses | 50 |

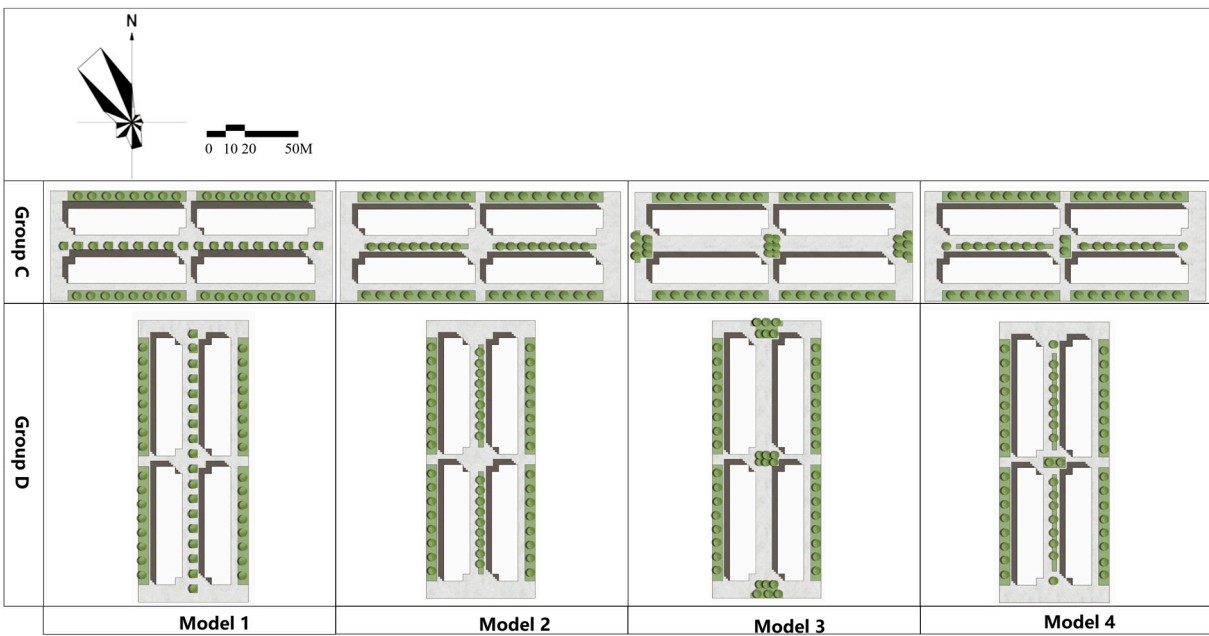

**Figure 8.** Design of experiment for greening layout.

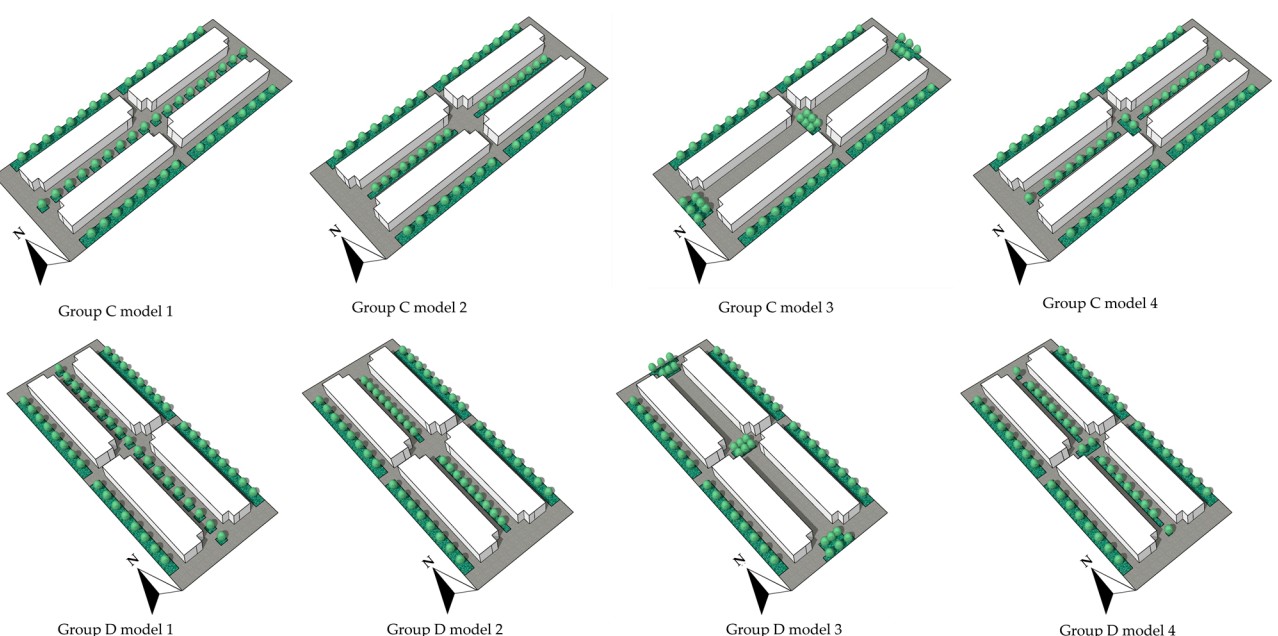

**Figure 9.** Three-dimensional diagram of each model in study 3.

## 3. Results and Discussions

### 3.1. Analysis on Experimental Results

3.1.1. Analysis of Microclimate Results of Experiments for Moving Lines of Streets

ENVI-met software was used to calculate the PET values for the east–west street perpendicular to the dominant wind direction (Model 1), the north–south street parallel to the dominant wind direction (Model 2), and the multi-line "cross"-shaped street (Model 3). At the study time of 2:00 p.m., a comparison was made among the microclimate environment characteristics of the three kinds of streets. It was calculated that the average PET value in the street perpendicular to the dominant wind direction in Model 1 was 55.13 °C; the average PET value in the street parallel to the dominant wind direction in Model 2 was 49.09 °C; and the average PET value in the "cross"-shaped street in Model 3 was 53.1 °C.

From the previous analysis on the corresponding relationship between PET and human senses (Figure 2), it is evident that in summer, the higher the PET value is, the hotter and the more uncomfortable the microclimate environment will be. Therefore, it can be concluded that the average comfort is the best for the street parallel to the dominant wind direction (Model 2), followed by the multi-line "cross"-shaped street (Model 3), and finally the street perpendicular to the dominant wind direction (Model 1).

Then, from an analysis of the calculated proportion of the PET values in each section of the street (Figure 10) and the distribution diagram of the PET values at 1.0 m (Figure 11), it is evident that when the street is a single-line type and when the entire street is perpendicular to the dominant wind direction (Model 1), the comfort of the microclimate is poor. When the street is a multi-line type, part of the street is perpendicular to the dominant wind direction and part of the street is parallel to the dominant wind direction (Model 3), the comfort of the microclimate is at a medium level. When the street is a single-line type and the entire street is parallel to the dominant wind direction (Model 2), the comfort of the microclimate is the best. This is because when cooling facilities (greening) are established at the entrance and inside of the street, the airflow entering the street has a cooling effect. The street parallel to the dominant wind direction can form a smooth ventilation corridor, and the microclimate environment within the entire street is relatively comfortable. When the street flow line is in a "cross" shape, only half of the street is in the unobstructed airflow, while the other half is shielded by buildings, resulting in slightly poorer airflow mobility. Therefore, the advantages and disadvantages of the microclimate environment are significantly impacted by spatial distribution. The microclimate environment is better in areas with unobstructed airflow, while it is worse in areas with unobstructed airflow. When the street is perpendicular to the dominant wind direction, there is no unobstructed airflow passage due to the shielding of buildings, so the microclimate environment in the street is poor. It can be concluded that maintaining a smooth airflow passage is very important for improving the microclimate environment of commercial streets.

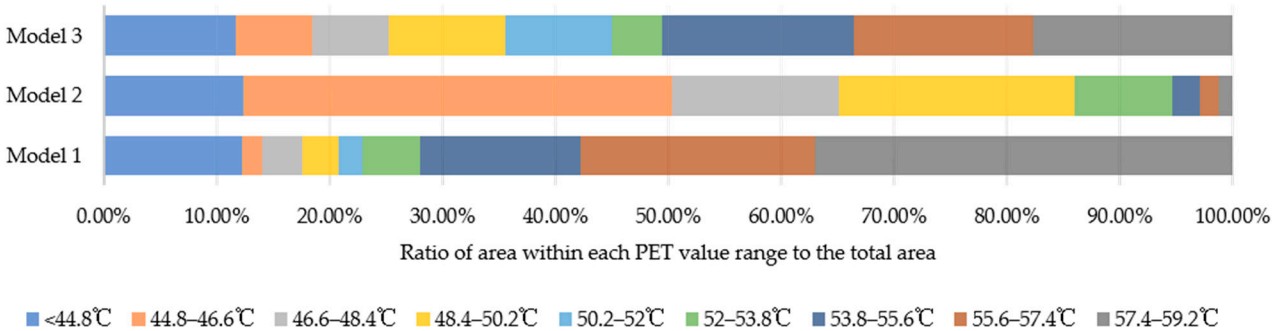

**Figure 10.** Distribution of PET values in different sections under different street flow lines.

### 3.1.2. Analysis of Microclimate Results of Experiments on Open Space Layout

When conducting the first group of experiments for the east–west street in Study 2, the average PET values in the streets under the three different open space layouts were calculated as follows: when the open space was located at both ends of the street (Model 1), the average PET value was 54.68 °C; when the open space was located in the middle of the street (Model 2), the average PET value was 54.44 °C; and when the open space was located at both the ends and the middle of the street (Model 3), the average PET value was 54.55 °C. There was no significant difference in the average comfort levels within the street. In the second group of experiments for the north–south street in Study 2, when the open space was located at both ends of the street (Model 1), the average PET value in the street was 50.39 °C; when the open space was located in the middle of the street (Model 2), the average PET value in the street was 51.84 °C; and when in a mixed situation, in which part of the open space was located at the end of the street and part of the open space was located in the middle of the street (Model 3), the average PET value in the street was 50.79 °C.

The overall average comfort was better when the open space was located at both ends of the street or in a mixed situation, and was worse when the open space was located in the middle of the street. Comparing Model 1 in Experiment 1 with Model 1 in Experiment 2, Model 2 in Experiment 1 with Model 2 in Experiment 2, and Model 3 in Experiment 1 with Model 3 in Experiment 2, the result shows that, under the same street layout, the internal microclimate environment of a street parallel to the dominant wind direction is better than that of a street perpendicular to the dominant wind direction, which is the same as the research results of the experiments conducted to assess the moving lines of the streets in Study 1.

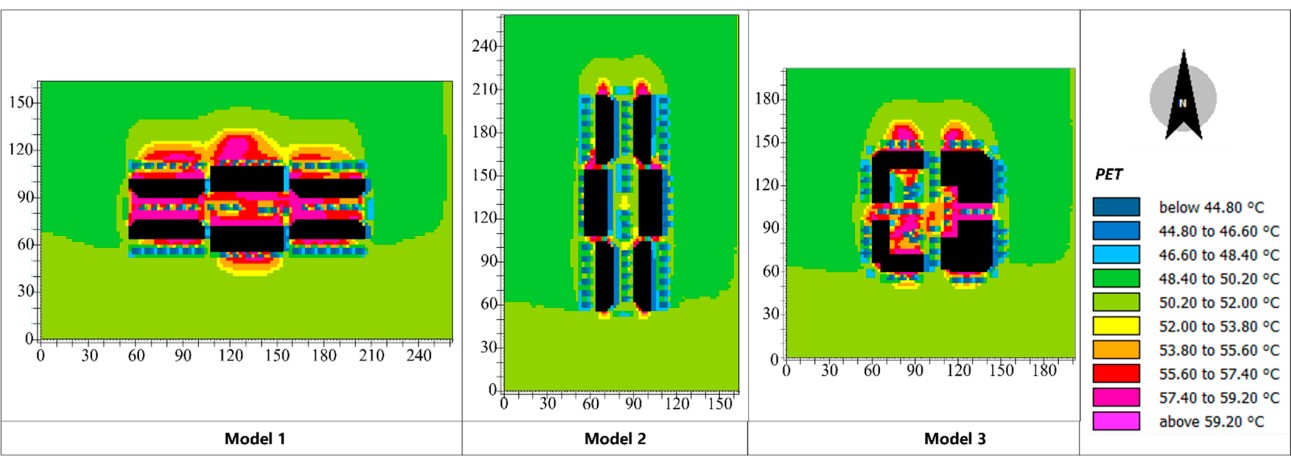

**Figure 11.** PET diagram at 1.0 m of the same street flow line.

According to the distribution of PET values in different sections of the east–west street with different open spaces in the first group of experiments (Figure 12) and the PET results at 1.0 m at 2:00 p.m. (Figure 13), when the open space was located in the middle of the street and at the ventilation corridor of the street, the microclimate environment of the open space is comfortable, and the larger the area of the open space located in the middle of the street is, the more comfortable PET value distribution area will be. For example, in Experiment 1, the open space located in the middle of the street in Model 2 was larger than that in Model 3, and the open space located in the middle of the street in Model 3 was larger than that in Model 1. The final PET distribution result in the street was that except for the small PET value in greening, and other areas with small PET values were distributed in the open space in the middle of the street; in other words, the microclimate environment in the middle of the street was relatively comfortable, the area with a comfortable microclimate in Model 2 was larger than that in Model 3, and the area with a comfortable microclimate in Model 3 was larger than that in Model 1. This is because the open space in the middle of the street was located on the air passage of the street, and smooth airflow could flow through the passage from south to north in summer, which is equivalent to setting up an air buffer area in the open space, forming a "wind storage" function in the open space. The larger the area is, the stronger the "wind storage" function and the more comfortable the microclimate environment will be.

According to the distribution of the PET values in different sections of the north–south street with different open spaces in the second group of experiments (Figure 14) and the PET results at 1.0 m at 2:00 p.m. (Figure 15), when the street was parallel to the dominant wind direction, the microclimate environment was comfortable when the open space was located at the ends of the street; and the larger the open space at the ends is, the more comfortable the microclimate environment of the street will be. For example, in Experiment 2, the open space at the ends of the street in Model 1 was larger than that in Model 3, and the open space at the ends of the street in Model 3 was larger than that in Model 2. The final PET distribution result in the street was that the comfort of the street space in Model 1 was better than that in Model 3, and the comfort of the street space in Model 3 was better

than that in Model 2. This is because an open space at the ends of the street is equivalent to opening an open space at the air outlet on the street. The larger the open space is, the more efficiently external wind is absorbed and the more comfortable the microclimate environment will be.

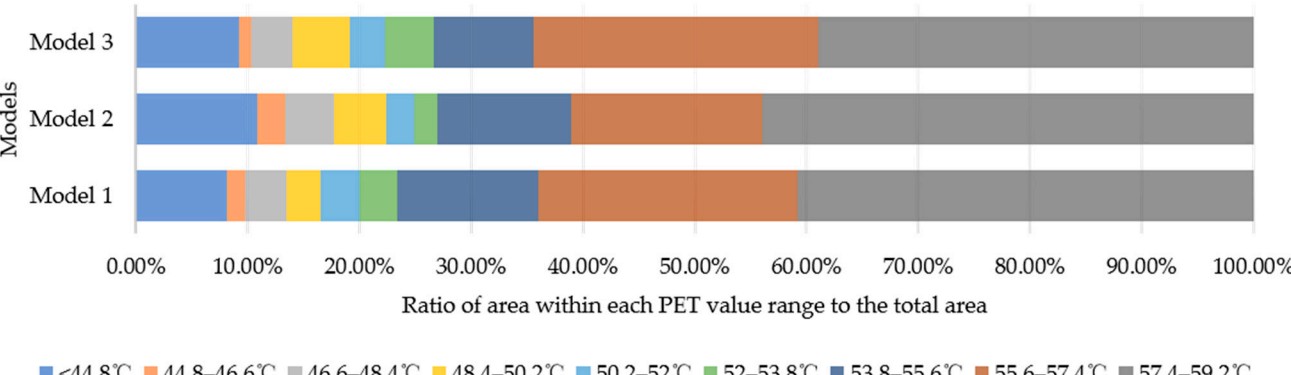

**Figure 12.** Distribution of PET values in sections with different open spaces on east–west street.

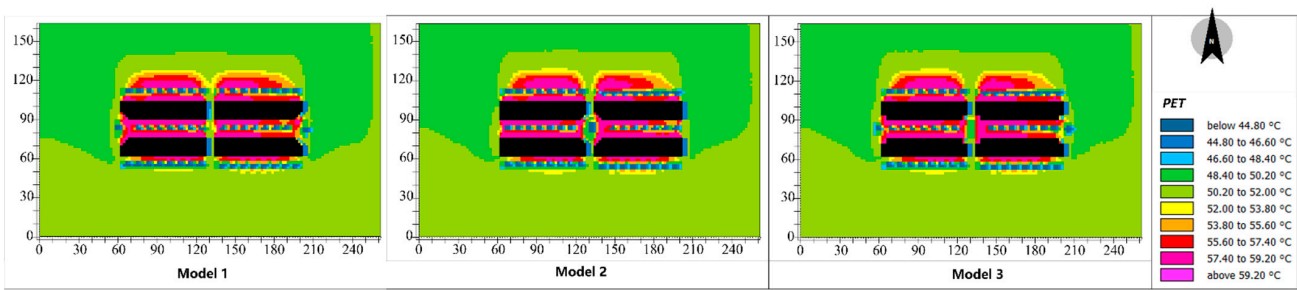

**Figure 13.** PET values at 1.0 m in different open spaces on east–west street.

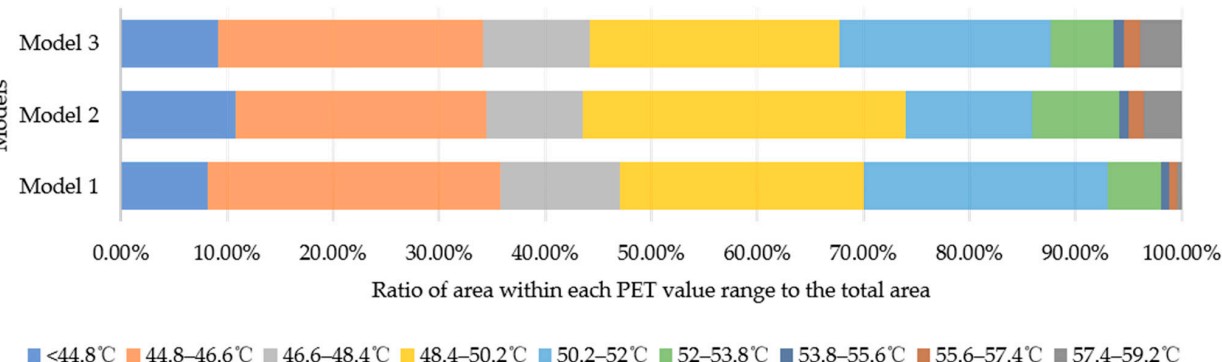

**Figure 14.** Distribution of PET values in sections with different open spaces on north–south street.

### 3.1.3. Analysis of Microclimate Results of Experiments on Greening Layout

In the first group of experiments in Study 3, when the greening layout of the east–west street was designed as point greening (Model 1), the average PET value in the street was 54.58 °C; when the greening layout was designed as linear greening (Model 2), the average PET value in the street was 54.31 °C; when the greening layout was designed as planar greening (Model 3), the average PET value in the street was 56.66 °C; and when the greening layout was designed as a mixed point, linear and planar greening (Model 4), the average PET value in the street was 54.46 °C. Under the same greening area, the microclimate environment in the street with only planar greening was poor. When the greening layout was designed as point greening, linear greening, or mixed point, linear and planar greening, the microclimate environment in the street was better than in the planar greening space and the average value difference was small. In the second group of experiments in Study

3, when the greening layout of the north–south street was designed as point greening (Model 1), the average PET value in the street was 51.16 °C; when the greening layout was designed as linear greening (Model 2), the average PET value in the street was 51.12 °C; when the greening layout was designed as planar greening (Model 3), the average PET value in the street was 50.79 °C; and when the greening layout was designed as mixed point, linear and planar greening (Model 4), the average PET value in the street was 50.78 °C. The difference in the average comfort within the street was small.

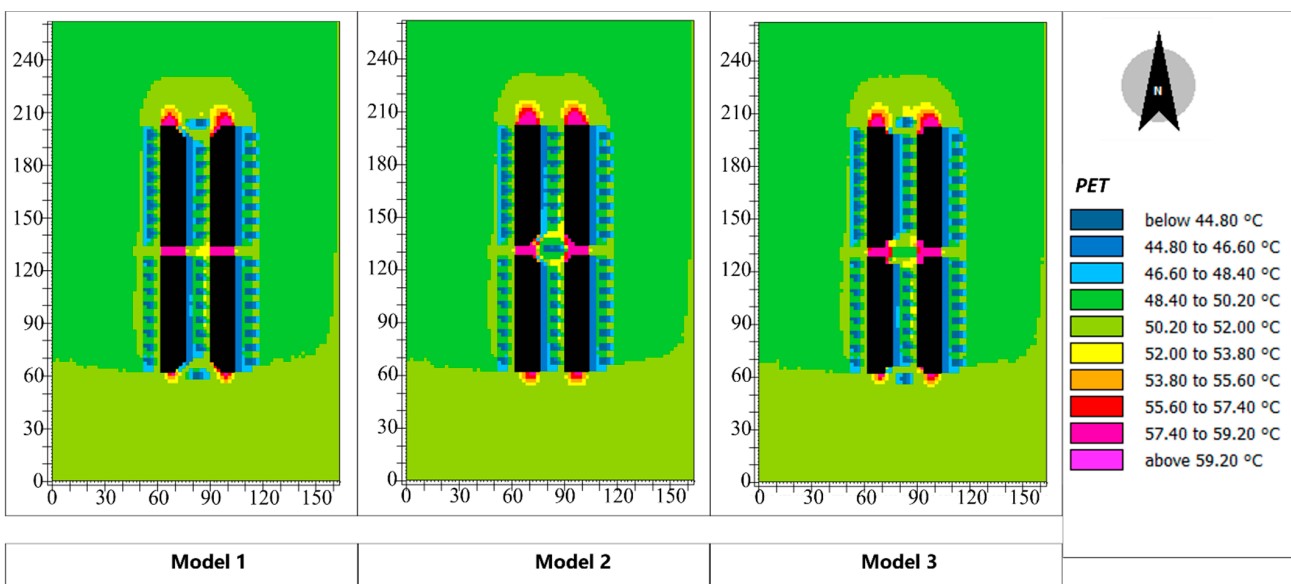

**Figure 15.** PET values at 1.0 m in different open spaces on north–south street.

According to the distribution of PET values in sections of the east–west street with different greening layouts (Figure 16) and the PET results at 1.0 m at 2:00 p.m. (Figure 17), under the same greening area, when the greening layout was centralized, the microclimate environment in the street was poor, and when the greening layout was designed as mixed point, linear and planar greening, the microclimate environment was better. This is because the original airflow in the east–west street was poor in circulation, and large-scale planting and greening further hindered the airflow, worsening the microclimate environment inside the street. In contrast, the use of dispersed point or linear layouts has a smaller hindering effect on the airflow, and the airflow generated by greening with a cooling effect circulates throughout the street, resulting in a better microclimate environment.

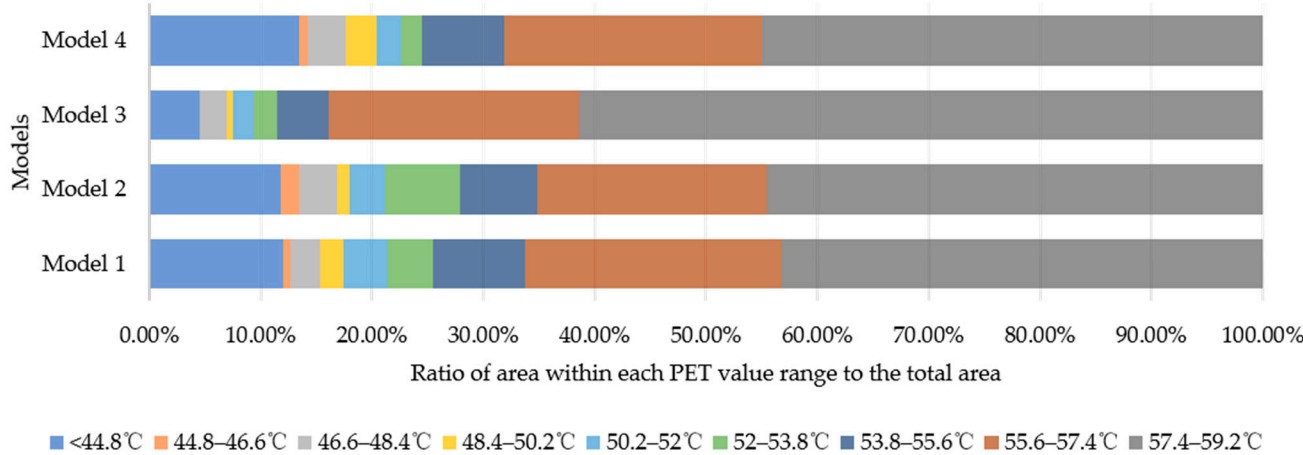

**Figure 16.** Distribution of PET values in sections with different greening layouts on east–west street.

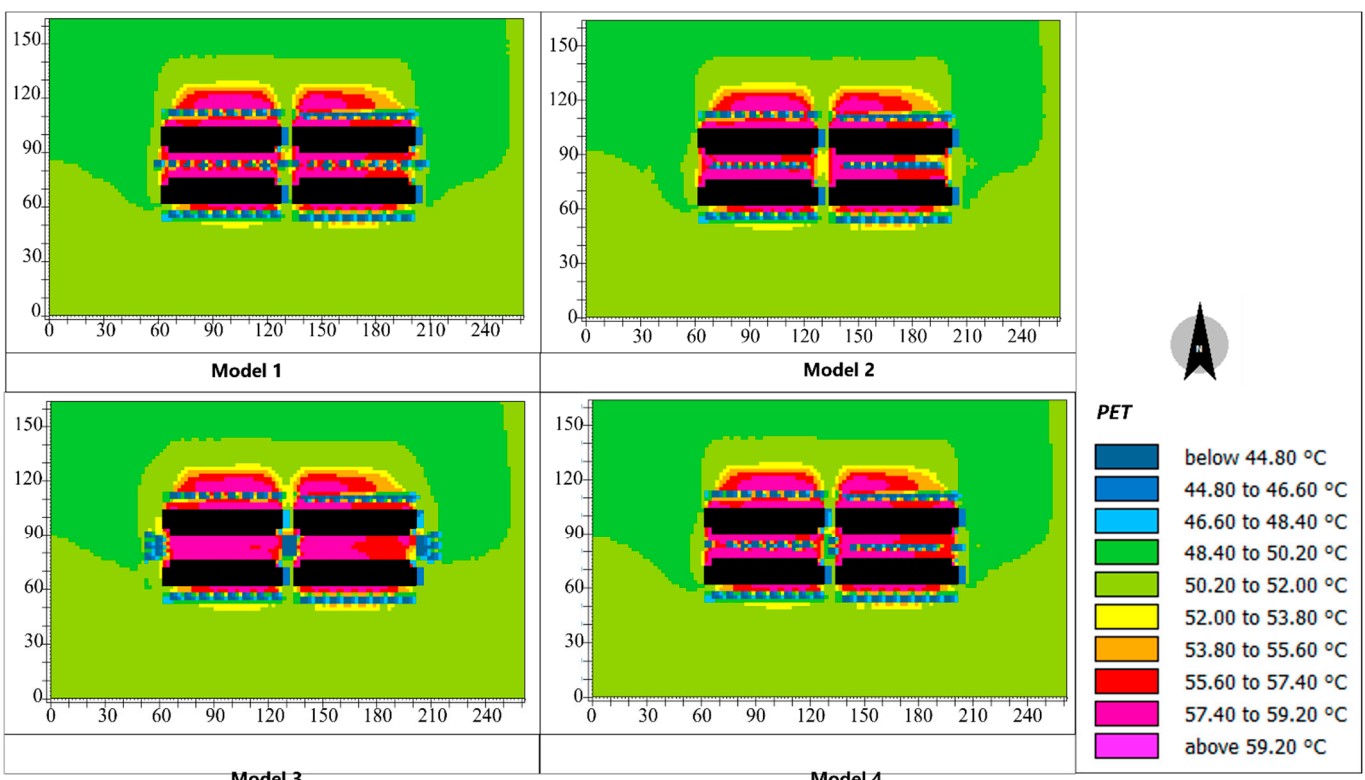

**Figure 17.** PET diagram at 1.0 m for different greening layouts on east–west street.

According to the distribution of PET values in sections of the north–south street with different greening layouts (Figure 18) and the PET results at 1.0 m at 2:00 p.m. (Figure 19), when adopting planar greening, although the proportion of sections with relatively small PET values is relatively small, the proportion of extreme sections with relatively large PET values is also small, and the average comfort of the entire street is better than the other three types of layouts. This is because the north–south street is parallel to the dominant wind direction. Under the limited greening area, the four greening layouts have a smaller hindering effect on the airflow entering the street, and the airflow in the street is unblocked. Arranging a large area of planar greening space in the upwind direction of the street can enable the external wind passing through the street to carry the low-temperature air generated by the upwind planar greening space, thereby improving the microclimate within the street in the absence of green plants inside the street.

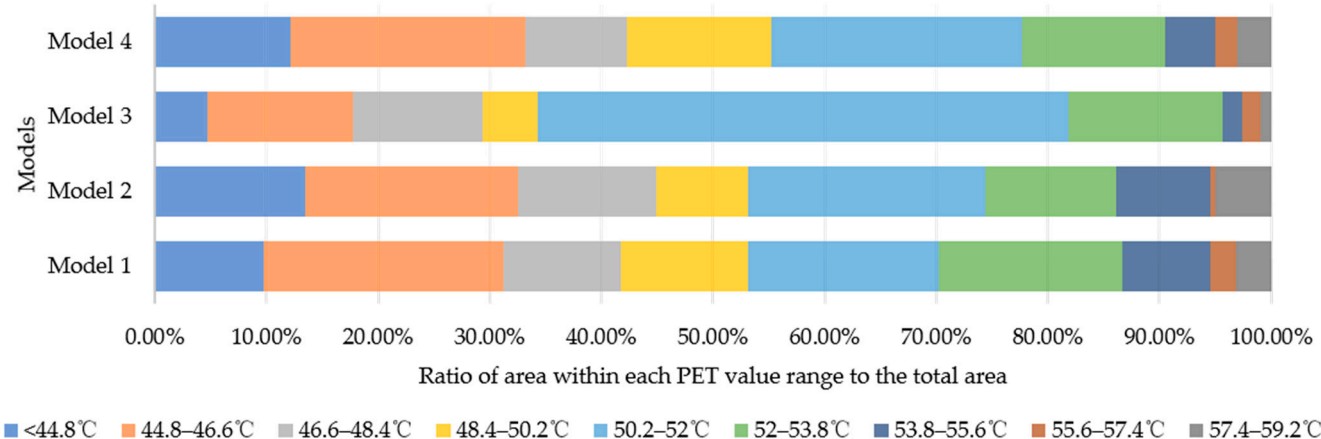

**Figure 18.** Distribution of PET values in sections with different greening layouts on north–south street.

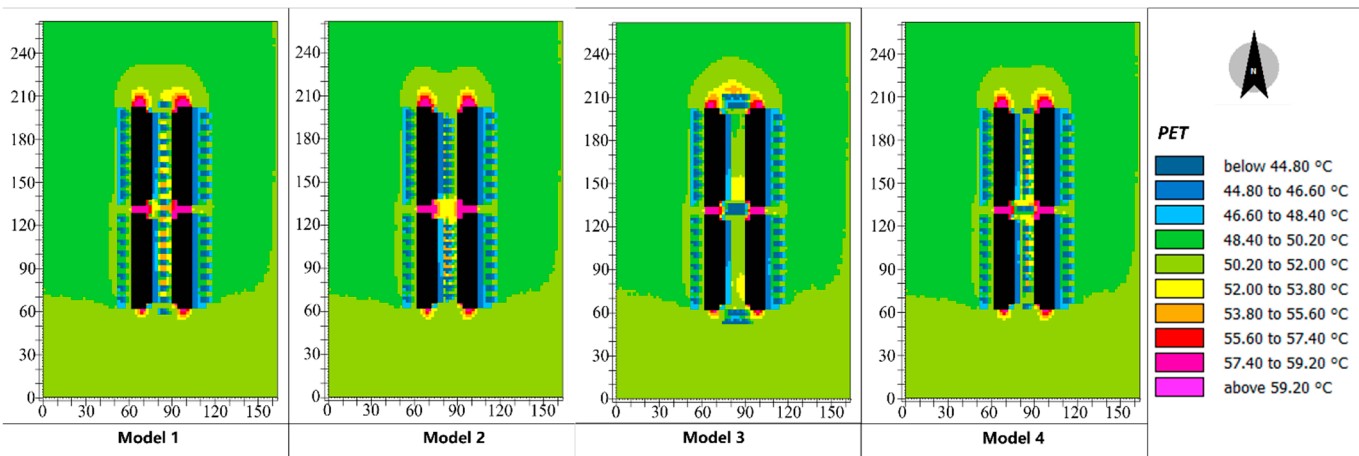

**Figure 19.** PET diagram at 1.0 m for different greening layouts on north-south street.

### *3.2. Optimization Strategies for Street Space of Pedestrian Commercial Street*

#### 3.2.1. Optimization Strategy for Street Flow Line Direction

According to the experimental results obtained for the street flow line in Experiment 1, it is very important to analyze the dominant wind direction of the city in the design of the street flow line. The microclimate environment inside the street parallel to the dominant wind direction of the city is significantly better than that inside the street perpendicular to the dominant wind direction. Therefore, in the design of a commercial street, it is necessary to reasonably design the street flow line according to the conditions of the plot and reserve a passage for dominant wind with appropriate width, so as to meet the use needs, facilitate the smooth passage of the dominant wind through the street in the current season, and optimize the microclimate environment inside the street.

#### 3.2.2. Optimization Strategy for Open Space of Street

In summer, in addition to the cooling effect of greening, factors that have a positive impact on the microclimate environment also include the improvement effect of the wind environment. Therefore, the wind environment should be given priority for the improvement of the microclimate in summer. In addition to reserving the potential air passage, the "wind suction" and "wind storage" functions of the street also greatly promote wind speed in the street. The research results of the experiment for open space in Experiment 2 show that using the potential air passage inside the street and the settings of the open space to build an air buffer area inside the street can increase the function of "wind storage", which can improve the comfort of the microclimate. Moreover, an open space can be built in the upwind position of the street to increase the "wind suction" function and better guide the dominant urban wind into the street.

#### 3.2.3. Optimization Strategy for Greening Layout of Street

According to the experimental results obtained for greening layout in Experiment 3, when the ventilation in the street is not smooth, the layout of the planar greening space should be minimized to prevent dense greening from hindering the wind flow and worsening the microclimate environment in the street. In this case, a green area can be decomposed into a multiple point greening space or a linear greening space, which can improve the poor microclimate environment. This approach can also expand the coverage of the improvement effect. When the airflow in the street is relatively smooth, planar greening space can be arranged in the upwind direction of the street. The larger the greening space area is, the more obvious the cooling effect will be. When planar green space is arranged in the upwind direction, the external wind can be utilized to convey the improved low-temperature air from the planar green space to the interior of the street, so as to expand the coverage of the cooling effect.

### 3.2.4. Thoughts on Winter Thermal Comfort in Pedestrian Commercial Streets

This research focuses on making improvements in the thermal comfort of pedestrian streets in summer in hot summer and cold winter regions. Another obvious feature of the hot summer and cold winter region is the severely low temperatures in winter. Some researchers have pointed out that plants can reduce the ambient temperature and make the ambient thermal comfort worse in winter [19,43–45]. In this case, a green space layout with a better cooling effect may have a more obvious negative impact on the thermal comfort of pedestrian malls. In contrast, some researchers have pointed out that plants also have the effect of improving environmental thermal comfort in winter [46–49]. This shows that although scholars generally believe that plants can help lower the temperature and improve the thermal comfort in summer [50–52], the impact of plants on the environmental thermal comfort in winter is more complex, and depends on a variety of different factors. Plants may have both positive and negative effects on the environmental thermal comfort. Compared with the impact of plants on the thermal comfort of the surrounding environment in different seasons, the impact of wind on the thermal comfort of the surrounding environment in different seasons is more uniform. Wind is generally considered to have a cooling effect in all seasons [53–57]. As for the issue of how to optimize the pedestrian commercial street in the hot summer and cold winter regions and enhance the thermal comfort, a conclusion cannot be drawn without a great number of studies. However, according to the viewpoints of relevant scholars, the ideal green space structure in winter can give full play to the role of green space and improve thermal comfort [46–49]; thus, optimizing the green space pattern of the pedestrian commercial street is entirely possible to further improve the thermal comfort in winter on the premise of having a good impact on thermal comfort in summer.

### 4. Conclusions

With the aim of reducing the negative impact of adverse outdoor microclimates on the volume of visitors to commercial streets, this paper adopts PET as the microclimate comfort index by taking into consideration the climatic conditions of Changsha, a typical city with a hot summer and cold winter, as well as the accuracy of assessing the outdoor microclimate environment and the difficulty of obtaining the indexes. Combining outdoor microclimate comfort with the techniques used to design commercial streets, alongside the premise of ensuring the functions of commercial streets, it is hoped that the revitalization of commercial streets through "micro transformation" will be realized and that some references for the future design of commercial streets are provided.

By analyzing typical pedestrian commercial streets in regions with hot summers and cold winters, the typical street patterns are reasonably derived. The outdoor microclimate conditions are simulated via the experimental method of controlling variables for the main morphological parameters of the typical street pattern (street flow line, open space, and greening layout) in order to compare the advantages and disadvantages of the microclimate environment, and the impact of street morphology in summer on the microclimate conditions is obtained. The results show that a smooth airflow corridor, an appropriate airflow passage shape, and a reasonable greening layout are the key to improving the microclimate environment of streets.

This study uses ENVI-met to explore ways in which to optimize the spatial structure of urban pedestrian commercial streets so as to improve their thermal comfort. ENVI-met is designed to essentially explore the highly complex interactions among urban architecture, vegetation and atmosphere, based on fluid mechanics and thermodynamic equations [58]. The relevant simulation results are representative of a certain kind of case. This study carries out an overall consideration of the spatial form characteristics of pedestrian commercial streets in the regions of China in which it is hot in the summer and cold in the winter, simulates patterns through the reasonable control of variables, and points out, according to the simulation results, that from the perspective of the street moving line, a ventilation corridor must be formed in order to improve the thermal comfort of these streets in summer according to the dominant wind direction. From the perspective of street open space, it is

necessary to form a larger air inlet in the dominant wind direction to improve the thermal comfort of pedestrian commercial streets. From the perspective of green plants, it is helpful to provide colder air to the pedestrian commercial streets by prioritizing the planning of larger areas of green plants at the air inlet in the prevailing wind direction in summer. In addition, priority should be given to planar green spaces in areas with good ventilation, and scattered green spaces in areas with poor ventilation. However, although the conclusions based on the relevant scenario models have guiding significance for the optimization of the spatial structure of pedestrian commercial streets, especially those in the regions of China in which it is hot in summer and cold in winter, the effects of the relevant strategies vary due to the different specific circumstances of the pedestrian commercial streets. Therefore, more research in this aspect is undoubtedly conducive to better understanding how to optimize the spatial structure of the regions in China in which it is hot in summer and cold in winter, thus improving thermal comfort in these regions. In addition, this study focuses on improving the summer thermal comfort of these regions by optimizing the spatial structure of their streets, but does not propose direct strategies regarding how to optimize the thermal comfort of pedestrian commercial streets in the winter. For this reason, a follow-up study on how to improve the spatial form of these regions in China, enhance their winter thermal comfort or to improve the spatial pattern of green space on the basis of this study, in order to raise thermal comfort of pedestrian commercial streets throughout the year, is undoubtedly of great practical significance.

**Author Contributions:** Conceptualization, J.L., H.T. and B.Z.; Methodology, J.L. and B.Z.; Software, J.L. and Z.S.; Validation, J.L. and H.T.; Formal analysis, J.L., Z.S. and H.T.; Investigation, H.T., J.L. and Z.S.; Resources, J.L. and H.T.; Data curation, J.L. and B.Z.; Writing—original draft, J.L., H.T. and B.Z.; Writing—review and editing, J.L., H.T. and Z.S.; Visualization, J.L. and B.Z.; Supervision, B.Z.; Project administration, J.L. and H.T. All authors have read and agreed to the published version of the manuscript.

**Funding:** This research received no external funding.

**Institutional Review Board Statement:** Not applicable.

**Informed Consent Statement:** Informed consent was obtained from all subjects involved in the study.

**Data Availability Statement:** Not applicable.

**Conflicts of Interest:** The authors declare no conflict of interest.

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
