# Peer review of "A Study on the Summer Microclimate Environment of Public Space and Pedestrian Commercial Streets in Regions with Hot Summers and Cold Winters"

_applsci, doi:10.3390/app13095263_

Round 1
Reviewer 1 Report
This article addresses the current issue of microclimate quality in urban public spaces. The authors attempted to compare the quality of microclimatic comfort in different street models in a climate with hot summers and cold winters. The chosen method of research, is not original, but the authors deliberately chose it over others as the most appropriate for the purpose of the study. The 3 street models were compared with regard to 3 different criteria (position relative to the wind, shape of the space and layout of greenery). The research required a limitation on the number of models tested, and this caused difficulty in choosing the most appropriate for the quality of the conclusions. The authors justified the choice of models well, although it can be debated whether this is not too small a number to be able to create indications for a programme of upgrading the space of specific cities (such was the authors' stated aim).
The overall methodology adopted is logically justified and the research has been carried out carefully. The authors described them well and drew conclusions in line with the research objectives undertaken. However, I have concerns about the following points:
1. The research was limited to the summer period due to the greater severity of this period compared to other seasons. However, it would have been appropriate to refer to these as well, if only in the discussion. This is particularly the case in winter, where the aims of optimising the microclimate are the opposite of those in summer, e.g. protection from wind is advisable.
2 No reference was made to the morphology of the urban development. The authors announced this in the introduction, but did not justify this decision. The height of buildings in the models studied is not without influence on the microclimate, so this issue should have been clarified and well justified in the introduction.
In addition to these fundamental concerns, I also have comments on the description of the research. The drawings of the models are not very understandable. They should be supplemented, for example, with three-dimensional models that clearly show where the buildings are and where the open space is. The concluding section, which lacks elements of discussion, also needs to be supplemented. Reference should be made to other seasons, to the question of the morphology of the buildings and to the lines of research to be undertaken in the future.
A minor comment is that the methods section mistakenly includes text from the instructions to authors.
Author Response
Suggestion 1: This article addresses the current issue of microclimate quality in urban public spaces. The authors attempted to compare the quality of microclimatic comfort in different street models in a climate with hot summers and cold winters. The chosen method of research, is not original, but the authors deliberately chose it over others as the most appropriate for the purpose of the study. The 3 street models were compared with regard to 3 different criteria (position relative to the wind, shape of the space and layout of greenery). The research required a limitation on the number of models tested, and this caused difficulty in choosing the most appropriate for the quality of the conclusions. The authors justified the choice of models well, although it can be debated whether this is not too small a number to be able to create indications for a programme of upgrading the space of specific cities (such was the authors' stated aim).
Comment 1: Thank you for your valuable suggestion. We have made comprehensive revisions to the article based on your suggestions. We have added a paragraph at the end of the conclusion part of the article to discuss the limitations of this study and whether and to what extent the results of this study have guiding significance. In that paragraph, we first stated that ENVI-met explores highly complex interactions between urban architecture, vegetation and atmosphere according to fluid dynamics and thermodynamic equations. The results obtained based on the relevant equations undoubtedly represent a kind of situation. Besides, we focus on exploring optimization strategies from the perspectives of the street moving line, extended space, and green space structure. Its essence is how to optimize the wind environment and green space pattern to improve the thermal comfort of commercial pedestrian streets in summer. The measures such as promoting the ventilation of ventilation corridors according to the prevailing wind in summer, making big openings to promote the wind to enter the ventilation corridor, setting the green space as a cooling facility at the air outlet to improve the comfort of the wind environment, setting a large area of green space in the middle of the pedestrian commercial streets is conducive to store wind and improve comfort, and Priority should be given to planar green spaces in areas with good ventilation, and scattered green spaces in areas with poor ventilation undoubtedly have guiding significance for improving the thermal comfort of commercial pedestrian streets in regions of China with hot summer and cold winter. We have also pointed out that due to the different specific conditions of different commercial pedestrian streets, the effects of relevant optimization strategies will also be different. We believe that more relevant studies can be conducted to enrich such results and better guide urban planners to optimize the green space patterns of commercial pedestrian districts to improve the thermal comfort in commercial pedestrian streets.
Suggestion 2: The overall methodology adopted is logically justified and the research has been carried out carefully. The authors described them well and drew conclusions in line with the research objectives undertaken. However, I have concerns about the following points:
Comment 2: Thank you very much for your recognition. We have carefully revised the article according to related guiding opinions. We hope to publish this article under your guidance.
Suggestion 3: 1. The research was limited to the summer period due to the greater severity of this period compared to other seasons. However, it would have been appropriate to refer to these as well, if only in the discussion. This is particularly the case in winter, where the aims of optimising the microclimate are the opposite of those in summer, e.g. protection from wind is advisable.
Comment 3: We have added a section 3.2.4 in Chapter 3. Results and Discussions. This section focuses on thoughts on Winter Thermal Comfort in Pedestrian Commercial Streets. Based on a large number of research literature, we point out that from the wind point of view, better ventilation configuration may be detrimental to the thermal comfort of pedestrian commercial streets in winter. However, some researchers have pointed out that the thermal comfort of pedestrian commercial streets can be improved by good green space layout in winter. This research focuses on optimizing the spatial pattern of pedestrian commercial streets in summer to improve thermal comfort. Follow-up researchers can take this as a basis and explore how to optimize the spatial pattern of green space layout in pedestrian commercial streets in order to make it possible to play a good role in improving thermal comfort in both winter and summer. If the green space can not enhance the thermal comfort in both winter and summer, how to make the most favorable choice for improving the thermal comfort of pedestrian commercial streets also needs more research to demonstrate.
Suggestion 4: 2 No reference was made to the morphology of the urban development. The authors announced this in the introduction, but did not justify this decision. The height of buildings in the models studied is not without influence on the microclimate, so this issue should have been clarified and well justified in the introduction.
Comment 4: The morphology of the urban development is added in the introduction part (please see the second paragraph of the introduction). As the paper mainly studies the pedestrian commercial streets, the literature on the morphological characteristics and changing process of pedestrian commercial streets are combed out. By summarizing relevant literature, it reveals that the characteristics of pedestrian commercial streets have certain commonalities. In addition, Chinese cities attach importance to the protection of the texture of pedestrian commercial streets in the process of development, which also lays a foundation for us to study the classification of pedestrian commercial streets later. In addition, in the fourth paragraph of the introduction section, we introduced the impact of different building heights on the microclimate of pedestrian commercial streets. The finding is the thermal comfort of the street can be improved by blocking sunlight with architectural shadow during the day in summer for buildings on both sides of the road. The wider the coverage of the architectural shadow is, the wider the range of thermal comfort improved by it is. Our subsequent analysis is also closely related to this finding.
Suggestion 5: In addition to these fundamental concerns, I also have comments on the description of the research. The drawings of the models are not very understandable. They should be supplemented, for example, with three-dimensional models that clearly show where the buildings are and where the open space is. The concluding section, which lacks elements of discussion, also needs to be supplemented. Reference should be made to other seasons, to the question of the morphology of the buildings and to the lines of research to be undertaken in the future.
Comment 5: We are sorry. We have added figures of three-dimensional models for each model. We think the new figures 5、7、9 can clearly show where the buildings are and where the open space . We have read some articles related to the thermal comfort of pedestrian streets in winter and added the relevant findings in the newly added section 3.2.4 in section 3 Results and Discussions. We have added discussion in the conclusion part.
Suggestion 6: A minor comment is that the methods section mistakenly includes text from the instructions to authors.
Comment 6: We apologize for this. We will carefully check and avoid such errors due to negligence after typesetting in the future.
Reviewer 2 Report
The research brings important contributions insofar as it aims to offer subsidies to the external microclimate in commercial streets, aiming at "micro renovation" and providing some reference for future projects of commercial streets.
In the item “Featured application”, it is stated, “The work has the potential to guide the design of pedestrian shopping streets to mitigate the negative impact of urban heat islands”. It is suggested to change this item, as what the article presents refers to the characteristics of the regional climate (hot summer and cold winter) and not exactly the effect of the heat island in these areas. The theme “heat island” does not appear in other parts of the text.
It is suggested that the introduction further explore the research carried out on the subject (outdoor microclimate) and on the area chosen for study. It is important that the research “problem” is widely highlighted.
The cited references are important, however, it is appropriate to expand the references on the subject and the area of study.
In general, the figures are well presented; however, I suggest some improvements. In the figure 1, it is necessary to insert the basic cartographic conventions of the maps, such as the geographic coordinates, the scale, and the captions. On the x and y axes of figures 7, 9, 12, 13, and 15, it is important to include information about what is being represented (measurement unit).
The methods are adequately described and the results are clear.
Author Response
Suggestion 1: The research brings important contributions insofar as it aims to offer subsidies to the external microclimate in commercial streets, aiming at "micro renovation" and providing some reference for future projects of commercial streets.
Comment 1: Thanks for your acknowledgment. We will try our best to improve the article and make it meet the publication requirement.
Suggestion 2: In the item “Featured application”, it is stated, “The work has the potential to guide the design of pedestrian shopping streets to mitigate the negative impact of urban heat islands”. It is suggested to change this item, as what the article presents refers to the characteristics of the regional climate (hot summer and cold winter) and not exactly the effect of the heat island in these areas. The theme “heat island” does not appear in other parts of the text.
Comment 2: Thanks for your suggestion. We have rewritten the featured application. The new one is ‘This study can guide urban planners and urban designers to think about how to improve thermal comfort by optimizing street flow lines, locations of open space, and greening arrangements in pedestrian commercial streets, especially in pedestrian commercial streets in hot summer and cold winter region in China.’
Suggestion 3: It is suggested that the introduction further explore the research carried out on the subject (outdoor microclimate) and on the area chosen for study. It is important that the research “problem” is widely highlighted.
Comment 3: Through an extensive literature review, the paper expounds the subject in the introduction section. In the third paragraph of the introduction, it further supplements the content of the original review, so as to better introduce the current research progress. In the fourth paragraph of introduction, the articles on improving the spatial form of blocks to improve their thermal comfort in hot summer and cold winter zones in China are sorted out. Through the review of relevant literature, it can be found that improving the spatial form of urban blocks to enhance the thermal comfort of urban blocks has drawn some researchers’ attention. The region in China with hot summers and cold winters has high temperatures in summer and coldness in winter, so it is very important to improve the thermal comfort of the living environment in this region. Therefore, the study on the spatial form of pedestrian commercial streets in China’s hot summer and cold winter region and the exploration of how to optimize commercial blocks from the perspective of spatial form will undoubtedly help improve the thermal comfort of pedestrian commercial blocks and increase their attractiveness. However, there are few relevant research on it. Considering the reality of pedestrian commercial streets in hot summer and cold winter region in China, the paper explores the spatial form improvement for thermal comfort of pedestrian commercial blocks from optimizing street flow lines, locations of open space, and greening arrangements in pedestrian commercial streets and greening arrangements in pedestrian commercial streets. It points out the importance of our research in the light of this situation.
Suggestion 4: The cited references are important, however, it is appropriate to expand the references on the subject and the area of study.
Comment 4: We are sorry. We have expanded the references on the subject and the area of study in the introduction. All cited references include the subject and the area of study.
Suggestion 4: In general, the figures are well presented; however, I suggest some improvements. In the figure 1, it is necessary to insert the basic cartographic conventions of the maps, such as the geographic coordinates, the scale, and the captions. On the x and y axes of figures 7, 9, 12, 13, and 15, it is important to include information about what is being represented (measurement unit).
Comment 4: We have revised these figures. We have added coordinates, the scale and the captions for figure 1. We have added information about what is being represented(measurement unit) in figures 7, 9, 12, 13 and 15.
Suggestion 5: The methods are adequately described and the results are clear.
Comment 5: Thanks for your acknowledgment. We will try our best to make the article meet the publishing requirement.